# Quality of Life, Family Support, Spirometry, and 6-Minute Walking Distance Differences between COVID-19 and Non-COVID-19 Intensive Care Unit Patients in One Year Following Hospital Discharge

**DOI:** 10.3390/healthcare12100996

**Published:** 2024-05-13

**Authors:** Konstantina Avgeri, Konstantinos Mantzarlis, Effrosyni Gerovasileiou, Konstantina Deskata, Maria Chatzi, George Fotakopoulos, Markos Sgantzos, Vasiliki Tsolaki, Epaminondas Zakynthinos, Demosthenes Makris

**Affiliations:** Medical Deparment, University of Thessaly, 41336 Larissa, Greece; augkwnna_2015@yahoo.gr (K.A.); gerovasileiou@yahoo.com (E.G.); konstadv@gmail.com (K.D.); mariahatzi1@gmail.com (M.C.); gfotakopoulos@med.uth.gr (G.F.); sgantzosmarkos@gmail.com (M.S.); vastsolaki@uth.gr (V.T.); ezakynth@med.uth.gr (E.Z.); appollon7@hotmail.com (D.M.)

**Keywords:** ICU, patients’ support, family support, quality of life, critical care, COVID-19

## Abstract

Background: Critically ill patients after Intensive Care Unit (ICU) discharge may present disability in their cognitive and physical functions. Objectives: To investigate the quality of life (QoL) of both COVID-19 and non-COVID-19 patients following ICU discharge, lung function, and physical performance of participants. Methods: This study was prospective and conducted between 2020 and 2021 in the “X” hospital. If patients were Mechanically-Ventilated (MV) > 48 h, they were included. Results: Fifty COVID-19 and seventy-two non-COVID-19 participants were included in this study. The mean (SD) of the total SF-36 scores at COVID-19 patients at hospital discharge and 3 and 12 months were 46.5 (14.5), 68.6 (17.8), and 82.3 (8.9) (*p* < 0.05), while non-COVID-19 participants were 48.5 (12.1), 72.2 (9.9), and 82.7 (5.4) (*p* < 0.05). The forced expiratory volume in one second (FEV1) and 6-minute walking distance (6MWD) were assessed at 3 and 12 months and significantly improved over 12 months. Conclusion: The QoL of COVID-19 patients improved significantly over time as FEV1 and 6MWD.

## 1. Introduction

Hospitalization in the Intensive Care Unit (ICU) is associated with increased rates of morbidity, causing physical, cognitive, and mental impairments, such as neuromuscular weakness and dysfunction, post-traumatic stress disorder, anxiety, and depression symptoms even years after discharge from the ICU [1,2]. Both patients and their families experience a highly stressful situation, presenting with reduced quality of life (QoL), as well as anxiety, depression, and post-traumatic stress disorder [1,3,4]. The majority of COVID-19 patients tend to experience a mild illness, presenting symptoms such as cough, shortness of breath, myalgia, and fatigue, while a small percentage develop a severe respiratory infection, which requires prolonged hospitalization and mechanical ventilation (MV) [5]. MV is considered the most common method for managing chronically ill patients and is the cornerstone of respiratory failure intervention. Despite the benefits it may have, it often causes many complications, leading to high rates of morbidity and mortality [6]. Evidence shows that the QoL of COVID-19 patients remains low after discharge, even after recovery. Factors associated with this are older age, gender, ICU admission, and prolonged use of MV [7]. However, patient survival rates have increased significantly [5]. Despite the increased survival rates, these patients present long-term morbidities and require prolonged hospitalization in the ICU [8]. Similarly, for non-COVID-19 patients, MV is associated with serious complications, while the frequency of these complications increases over time, such as ventilator-associated pneumonia and increased mortality rates [9]. Differences are shown between COVID-19 and non-COVID-19 patients. A logical explanation is that patients with COVID-19 suffered from a severe and acute infection that significantly reduced their QoL during their hospitalization; however, compared to non-COVID-19 patients, there is a greater improvement over time. Longer duration of MV in both COVID-19 and non-COVID-19 patients is associated with a worse prognosis [10]. Despite the high survival rates of patients after acute respiratory failure and the use of MV in the ICU, it was observed that approximately 80% of patients experienced new or worsening physical, cognitive, and mental health disorders that persisted even after discharge from the ICU [11]. Patients presented both pulmonary and functional capacity six months after discharge from the ICU, whereas patients presented improvement 12 months after discharge [2], compromising patients’ QoL [11,12]. The role of the family is vital, as they undertake the quality care of patients and are responsible for making decisions, contributing to the improvement in their health status. The literature data present that a high percentage of patients need to be cared for by family members [13,14,15]. In this regard, the long-term impact of critical illness on the QoL of both COVID-19 and non-COVID-19 patients and family members is important for the design of effective supportive healthcare networks. However, data on the impact of family support after ICU, particularly in Greece, are limited.

In this study, we aimed to investigate the QoL of ICU patients with both COVID-19 and non-COVID-19 after ICU hospitalization, and we assessed the support of family members in their daily activities on their QoL at one-year follow-up. Moreover, we aimed to evaluate the physical performance and the lung function changes at a one-year follow-up.

## 2. Methods

This prospective study was conducted between 2019 and 2021 in the ICU at the University Hospital of Larissa, Greece. Patients were divided into two groups: COVID-19 and non-COVID-19; and medical and surgical patients. Inclusion criteria were (a) use of MV for more than 48 h, (b) ability to perform spirometry and 6-minute walking distance (6MWD) at discharge and during 3-month and 12-month follow-up, and (c) written informed consent from the patient to be interviewed and complete questionnaire to assess the QoL and family support in their daily activities.

This study was approved by the local ethics committee of the University Hospital of Larissa (No. 43704). Written informed consent was obtained by the patient or next of kin.

### 2.1. Study Outcomes

The primary outcome was Short Form-36 (SF-36) score change at a 12-month follow-up, as well as change in forced expiratory volume in one second (FEV1) and forced vital capacity (FVC) over time for COVID-19 patients compared to non-COVID-19 patients. Secondary outcomes were the daily support (hours/day) during COVID-19 and the support surgical patients received from their friends and family members.

### 2.2. Data Collection

The evaluation of medical and surgical patients and COVID-19 patients was performed at discharge, 3 and 12 months of follow-up, while spirometry and 6MWD of COVID-19 patients were performed at 3 and 12 months. Clinical assessment, Acute Physiology and Chronic Health Evaluation (APACHE II) severity of critical illness, ICU and hospital length of stay, spirometry, 6MWD, and QoL were included in the assessment of study patients. The APACHE II is a measure that assesses disease severity based on the patient’s current physiological measurements, age, and past health conditions. The score ranges from 0 to 71, with a higher score associated with an increased risk of in-hospital death [16].

The decision for admission and discharge of the patients was left to the attending physician.

### 2.3. Questionnaire Interview

The questionnaire interview was conducted using the SF-36 score questionnaire, which assessed the participants’ QoL, as well as a special questionnaire that assessed family support. The special questionnaire includes questions related to family and marital support as well as the support they received from their friends. All data were adjusted to the needs and hours of care for patients [13,17]. Classification included 1–4 h/day, 4–8 h/day, 24 h/day, and never. However, regarding the visits they received from their friends, the classification was arbitrarily carried out to every day, 3–4 times/week, once/week, 1–2 times/month, and never. The SF-36 score questionnaire includes multi-item scales that measure the eight health-related concepts: physical functioning (PF); role limitations due to physical health problems (RP); bodily pain (BP); general perceptions of health (GH); vitality (VT); increased energy levels and fatigue; social functioning (SF); role limitations due to emotional problems (RE); and mental health (MH). Each item is weighted on an additive scale to calculate the final domain score. A higher score indicates less impairment, and a lower score indicates more significant impairment. Each dimension is scored on a scale from 0 to 100. The specific questionnaire applies to the Greek population. A classification was made between COVID-19 and non-COVID-19 and medical and surgical patients, according to the median SF-36 score. This classification was made arbitrarily and defined as >median patients with an improved score and <median patients with an unimproved score. An appointment was scheduled with the participants at the hospital’s outpatient clinic, where the interview, the filling in of the questionnaire, the spirometry, and the 6MWD took place. For patients who were unable to attend the hospital, the appointment was scheduled at their home. In cases where the participant was not able to complete the questionnaire alone, the questions were answered with the assistance of the next of kin.

### 2.4. Respiratory Function Assessment and 6-Minute Walking Distance (6MWD)

Patients’ lung function was assessed by FEV1 and FVC measurements. The spirometry evaluation was planned and performed at discharge and at 3-month and 12-month follow-ups of the patients, using a portable spirometer, MIR Spirodoc. The spirometer includes disposable turbines. Each patient has spirometry performed in a standing position and has been instructed to close their lips tightly over the mouthpiece, use a flexible nasal caliper, and inhale and exhale normally several times. Each patient is then instructed to slowly inhale as deeply as possible, followed by a forceful exhalation. Spirometry was repeated 3 times, and the highest predicted value was recorded. Before starting and at the end of spirometry, saturation was checked with a pulse oximeter (SP02), which recorded the spirometry values using pulse oximetry (Nonin 8500 M, Nonin Medical, Minneapolis, MN, USA).

Moreover, patients were asked to perform 6MWD at the hospital’s outpatient clinic during their scheduled appointment. The 6MWD is used to assess aerobic capacity and endurance. The distance covered over a time of 6 min is used as the outcome by which to compare changes in performance capacity. It was conducted on a level, closed corridor, approximately 100 m. feet, which was rarely used. The assessment of the patients’ walking was 30 m in length and was marked every 3 m. All necessary instructions were given to patients according to accepted recommendations. The necessary equipment needed to walk was the stopwatch, the measuring wheel to measure the distance covered, an unobstructed path of 30 m, two cones to mark the distance to be covered, the pulse oximeter to measure heart rate, and SpO2 and a Borg dyspnea scale. The patient had to wear clothes and shoes that were comfortable and suitable for walking. Blood pressure, oxygen saturation, pulse oxygen, and baseline dyspnea were checked before the start and at the end of the examination using the Borg scale. The patient had to remain seated and at rest in a chair for at least 10 min before the start of the examination while being accompanied by a physician, who was present at the starting point of the patient’s walk. The patient was encouraged throughout the examination.

### 2.5. Statistical Analysis

Data were expressed as n (%) for categorical variables and mean ± SD (standard deviation) or median ± IQR (interquartile range, 25th, 75th quartiles) for continuous variables. Continuous variables between two groups were compared by a Mann–Whitney U test; categorical variables were compared using a chi-square test or Fisher’s exact test, where appropriateness and normality were assessed by the Shapiro–Wilcoxon test. All statistical tests were 2-sided. SPSS v.26.0 (IBM) software was used, considering *p*  <  0.05 statistically significant (ILLINOIS, USA).

## 3. Results

Overall, 50 COVID-19 patients and 72 non-COVID-19 patients were included in this study (Figure 1). The baseline clinical characteristics of participants are shown in Table 1.

Among the three patient groups (COVID-19, non-COVID-19 medical patients, and non-COVID-19 surgical patients), there was a statistically significant difference in the duration of ICU hospitalization (*p* = 0.01). There were also differences in terms of the post-ICU spirometric values between COVID-19 and non-COVID-19 patients for both FEV1 and FVC, *p* = 0.001 and *p* = 0.02, respectively, and between COVID-19 and surgical patients (for both FEV1 and FVC, *p* = 0.001 and *p* = 0.01, respectively). Α statistically significant difference was also shown in the 6-minute walking distance (6MWD) between COVID-19 and non-COVID-19 patients (*p* = 0.0001), between COVID-19 and medical patients (*p* = 0.0001), and between COVID-19 and surgical patients (*p* = 0.001).

### 3.1. Lung Function and 6MWT Post ICU

Spirometric values and 6MWD distance over time are presented in Figure 2a–c. FEV1 and FVC were significantly improved over the time points assessed in COVID-19 patients (*p* = 0.006 and *p* = 0.008) and non-COVID-19 patients (including separate assessments in medical and surgical groups); the changes between the time points of lung function evaluation were not significant. 

There was a significant improvement over time in the 6MWD distance in both COVID-19 and non-COVID-19 patients (medical and surgical) [*p* < 0.0001, *p* < 0.0001, and *p* < 0.0001, respectively]. The improvement was significantly greater in non-COVID-19 patients (*p* = 0.002). No statistically significant difference in 6MWD changes between medical and surgical patients was found.

### 3.2. Daily Support from Family and Friends

Details for daily support from family/friends at different time points are shown in Figure 3a–c. Figures show the frequency of support (hours/day) patients receive from their family members and spouses, as well as the visits/week they receive from their friends. Fourty-one out of fifty (82%) COVID-19 patients and thirty-six out of seventy-two (72%) non-COVID-19 patients were supported by two or more family members (*p* = 0.7), whereas forty-six (92%) COVID-19 patients and thirty-eight (76%) non-COVID-19 were supported by one or more friends (*p* = 0.5). No significant relationship was found between support (either by family or friends) and FEV1, FVC, 6MWD, days of hospitalization, or the type of admission. 

### 3.3. SF-36 Score Post ICU

Figure 4a–c presents the SF-36 total physical and mental health component scores of COVID-19 and non-COVID-19 (medical and surgical) participants over 12 months following hospital discharge. Figures show the statistically significant differences at different periods after ICU. There was no statistically significant difference between medical and surgical patients at baseline SF-36 total score. There was a significant improvement in the total SF-36 score in both groups (COVID-19 and surgical patients); improvement was greater in COVID-19 patients both at 3 months (*p* = 0.0001) and at 12 months of follow-up (*p* < 0.0001). Similar results were obtained for the physical and mental domains of the SF-36 score (*p* < 0.0001).

We further analyzed whether patients who presented SF-36 total score greater than the median score of the cohort over 12 months post ICU also presented specific characteristics in terms of lung function or 6MWD changes over time. We found that COVID-19 patients with SF-36 score > median score also presented a significant increase in FEV and FVC (*p* < 0.05), whereas there was a trend toward increased 6MWD values (*p* = 0.06). No significant association between lung function values, 6MWD, and SF-36 score changes were found in non-COVID-19 patients.

## 4. Discussion

The main findings from the present longitudinal study were that the SF-36 score improved in both COVID-19 and non-COVID-19 patients post ICU; however, the SF-36 score improvement was greater in COVID-19 patients. Moreover, we found out that the COVID-19 patients presented significant FEV1 and FVC changes over one year following the ICU discharge, whereas the changes in non-COVID-19 patients were not significant. Both COVID-19 and non-COVID-19 patients presented improved scores and 6MWD over time, although the improvement was greater in non-COVID-19 patients. When data were analyzed using the SF-36 median score, we found that COVID-19 patients with SF improvement > median score change presented statistically significant improvement in FEV1 and FVC (*p* < 0.05), whereas there was a trend toward increased 6MWD values (*p* = 0.06) compared to the rest of the cohort.

Patients admitted to the ICU with severe or mild symptoms present with a multitude of problems, manifesting reduced QoL compared to the general population, even years after admission [18,19,20]. Patients with COVID-19 present symptoms, such as fatigue, shortness of breath, memory loss, concentration, and sleep disorders, which persist long after their hospitalization in the ICU [21]. Previous studies showed [22,23,24,25] that patients surviving ICU may present reduced physical and social functioning even years following their discharge. The physical function of these patients remains reduced compared to the values of the patients before their admission to the ICU, as well as compared to the general population of the same age [26]. Similar are the results from the study of Deana et al. (2023) [27]. The median score of the mental and physical components is at moderate levels. While comparing between PTSD and non-PTSD patients, non-PTSD patients show better results. So, we conclude that patients admitted to the ICU with more serious problems present a reduced QoL. The present study shows that the SF-36 score of COVID-19 and non-COVID-19 patients presented improvement between discharge and 12-month follow-up. However, COVID-19 patients showed greater improvement compared to non-COVID-19 patients. A reasonable explanation is that COVID-19 is an acute and severe disease that may cause greater reductions in QoL during illness and greater improvement over time.

In line with the above, our findings suggest that patients with COVID-19 showed significant changes in FEV1 and FVC over one year following ICU. Obviously, in patients with COVID-19, respiratory function is significantly affected during the acute phase, where ICU admission may occur. Previous studies showed that the pulmonary function of patients is compromised [18,28,29] and is marked by reduced lung capacity and volume and respiratory muscle weakness [18,30]; however, there are gradual improvements in FEV1 and FVC [18,31,32,33], especially following interventions [34]. Hence, the lung functions of surviving patients with acute lung infections are expected to improve and may be normal in the majority of patients [35]. In our study, we have not found significant changes in lung functions in non-COVID-19 patients, and this might reflect that the majority of non-COVID-19 patients had no severe acute lung problems, which could acutely compromise their lung functions.

The present study shows improvement in the 6MWD in both groups of COVID-19 and non-COVID-19 for both medical and surgical patients; however, for the non-COVID-19 patients, improvement was most important. We found no association between 6MWD changes and patients’ characteristics, such as gender, age, or disease severity, which could potentially affect 6MWD performance. However, we acknowledge that we have not performed a detailed cardiovascular evaluation that might have depicted differences in 6MWD performance, and it is a limitation of our study. Previous studies [17] have shown that 6MWD 4 months after ICU discharge in COVID-19 patients is significantly impaired compared to healthy subjects [36,37]. Other studies have shown that exercise capacity may be impaired 1 year after discharge [28,29,38]. Nevertheless, similar to our results, an improvement in 6MWD is expected over time, and this may be associated with an improvement in QoL [18]. In our study, we also found that non-COVID-19 patients improved their 6MWD as well; this is in agreement with previous reports [39]. However, 6MWD improvement in non-COVID-19 patients was greater compared to COVID-19 patients. Since we found no specific parameter to be associated with the 6MWD change, we speculate that COVID-19 disease may have a long-term impact either on cardiovascular function or the musculoskeletal function in patients (considering that lung function improved significantly in COVID-19 patients over time). 

The present study presents certain limitations, which are important to consider when interpreting the results. First, there were some differences in the baseline characteristics of the participants, and this should be taken into account. Furthermore, the population studied might be relatively small, and this study was carried out in a single center in a Greek region. Therefore, the results from the present study can be considered more useful to implement strategies at the local level. Second, the questionnaire used did not include specific questions regarding patients’ health status before admission, details on private and public health support for patients, details about falls of patients, which is a major global health concern, and mental health support following ICU. In this respect, the impact of those variables on patients’ QoL cannot be assessed. Third, the present study does not provide details on mechanical ventilation-specific settings or the mechanical properties of the participants’ respiratory systems. We acknowledge that these details could have provided more insight into this investigation. Nevertheless, our findings may form the base of a larger future study that could assess QoL in those populations, taking into account the aforementioned variables as well. 

## 5. Conclusions

In conclusion, this observational study suggests significant improvement in QoL in COVID-19 and non-COVID-19 ICU patients over a year following ICU. We found no differences in SF-36 score changes between those two populations. We found differences between COVID-19 and non-COVID-19 ICU patients in terms of lung function and 6MWD over time. COVID-19 patients presented significant improvement in spirometric values over one year, whereas both groups presented improvement in 6MWD; the improvement was greater in COVID-19 patients. These differences may reflect the specific impact of COVID-19 on lung function and exercise performance in patients. A future study may provide more insight into the long-term impact of COVID-19 on the exercise performance of these patients.

## 6. Implications of Clinical Practice

The contributions of family members and friends are important for improving their QoL. Routine lung function assessment may be helpful and could be incorporated into the follow-up of critical care patients following ICU discharge. The involvement of family members could be incorporated into relevant programs aimed at improving the recovery of patients from critical illnesses.

## Figures and Tables

**Figure 1 healthcare-12-00996-f001:**
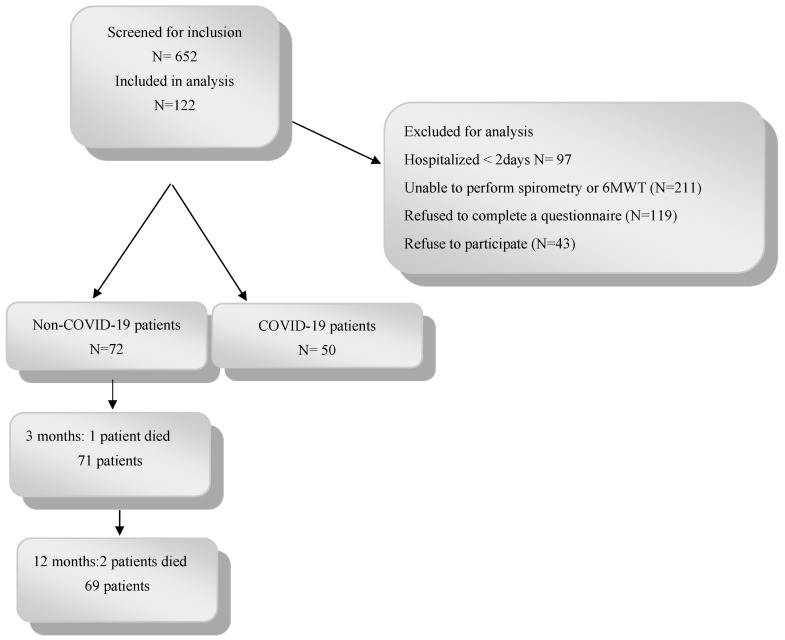
Show chart of this study.

**Figure 2 healthcare-12-00996-f002:**
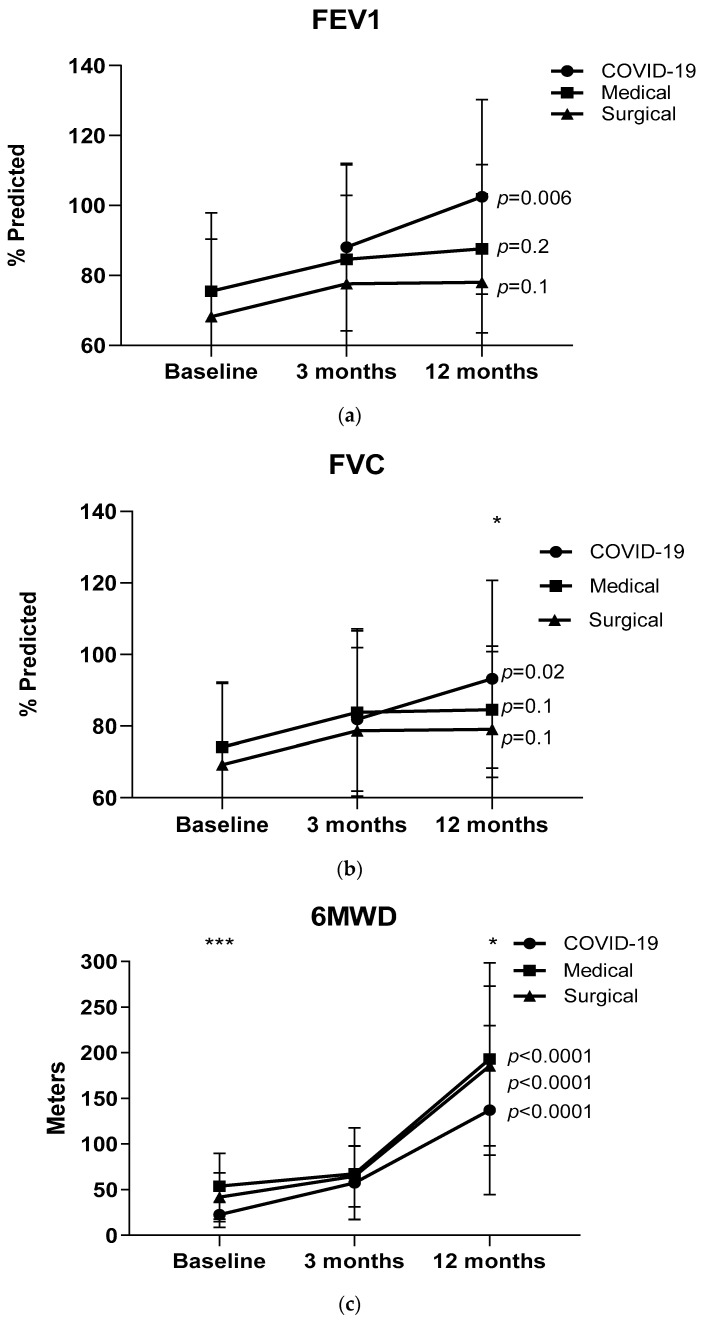
(**a**–**c**) Forced expiratory volume in one second (FEV1) and forced vital capacity (FVC) and the six-minute walking distance (6MWD) distances at 3 different periods after ICU discharge. Data are presented as mean (SD) values. * *p* < 0.05, *** *p* < 0.0001 difference between COVID-19 and medical and surgical.

**Figure 3 healthcare-12-00996-f003:**
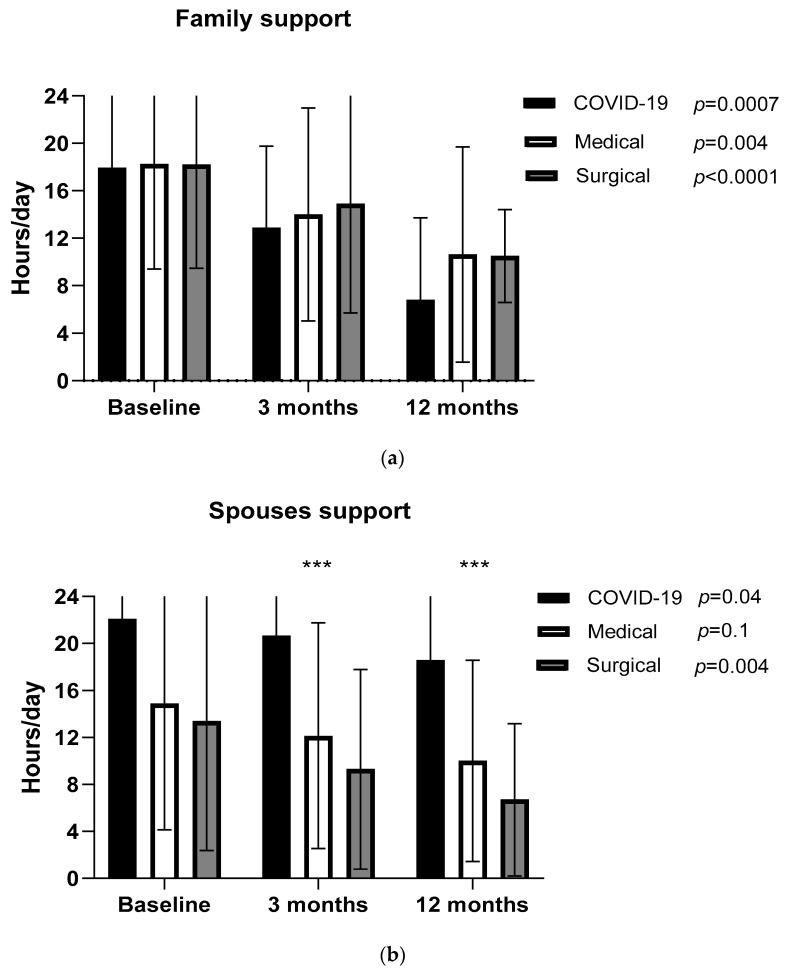
(**a**–**c**) Support from family members/spouses (hours/day) in daily activities and the visits by friends (frequency of visits/week) to patients at 3 different periods after ICU discharge. Data are presented as mean (SD) values. ** *p* < 0.001, *** *p* < 0.0001 difference between COVID-19 and medical and surgical.

**Figure 4 healthcare-12-00996-f004:**
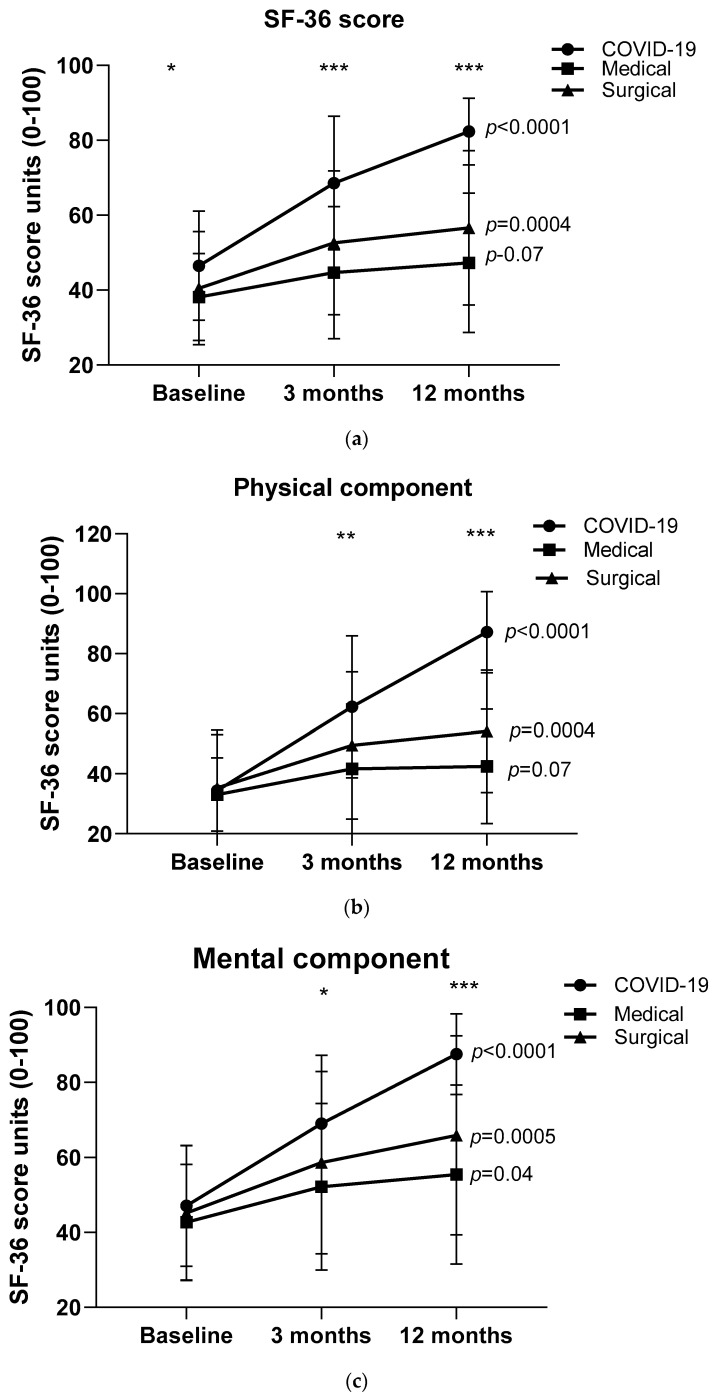
(**a**–**c**) SF-36 total score, physical and mental health scores of participants with and without COVID-19 (medical and surgical) at hospital discharge, 3-month, and 12-month follow-up. Data are presented as mean (SD) values. * *p* < 0.05, ** *p* < 0.001, *** *p* < 0.0001 difference between COVID-19 and medical and surgical.

**Table 1 healthcare-12-00996-t001:** Baseline characteristics of participants.

	COVID-19	All Non-COVID-19	Non-COVID-19 *p*-Value Medical Surgical		
	n = 50	n = 72	n = 32	n = 40	
Age, *	57.9 (13.7)	56.5 (17.7)	61.2 (15.2)	52.6 (18.7)	NS
Sex, Female, n (%)	25 (50)	25 (34.7)	11 (34.4)	14 (35)	NS
Comorbidities	32 (64)	51 (70.8)	24 (75)	27 (67.5)	NS
APACHE II score [median (IQR)], **, ***	18 (11.0–25.5)	14 (14.0–21.0)	21.0 (12.0–26.0)	14 (10.5–23.0)	*p* = 0.009
ARDS, *, **, ***	50 (100)	2 (2.8)	2 (6.25)	0 (0)	*p* < 0.0001
Mechanical ventilation duration, [median (IQR)], days	12.5 (6.5–17.5)	3.5 (3.5–6.0)	6.0 (3.0–12.5)	3.5 (2.5–9.0)	NS
ICU stay, [median (IQR)], days, *, ***	15.0 (9.0–21.5)	10.5 (10.4)	8.5 (3.25–14.7)	5.5 (3.0–11.0)	*p* = 0.009
Hospital stay, [median (IQR)], days	22.3(18.5–25.5)	11.0 (11.0–14.5)	14.5 (10.2–22.4)	11 (7.5–18.5)	NS
Time to return to daily routine, n (%)					*p* = 0.02
1–6 months	15 (30)	38 (52.8)	13 (40)	25 (62)	
6–12 months	21 (42)	9 (12.5)	5 (16)	4 (10)	
>12 months	19 (38)	25 (34.7)	14 (44)	11 (28)	
Rehabilitation after ICU, n (%)	10 (20)	30 (41.7)	16 (50)	14 (35)	NS
FEV1, % Pred, *, ***	88.1 (23.8)	70.9 (22.3)	75.5 (22.4)	68.2 (22.2)	NS
FVC, % Pred, *, ***	81.9 (20)	71.1 (21.0)	74.1 (18.2)	69.3 (22.6)	NS
FEV1/FVC, % Pred	108.5 (15.6)	104.2 (14.7)	105.5 (15.2)	103.3 (14.5)	NS
PEF, % Pred, *, **, ***	80.7 (29.6)	59.6 (23.8)	62.0 (23.0)	58.1 (24.5)	NS
6MWD, [mean (SD)], meters, *, **, ***	22.6 (13.5)	47.7 (31.5)	53.8 (36.0)	42.8 (27.0)	*p* < 0.0001

Data are presented as mean (SD) unless indicated otherwise. * *p* < 0.05 difference between COVID-19 and non-COVID-19. ** *p* < 0.05 difference between COVID-19 and medical. *** *p* < 0.05 difference between COVID-19 and surgical. ARDS: Acute respiratory distress syndrome; FEV1: Forced expiratory volume; FVC: Forced vital capacity; FEV1/FVC: the ratio of the forced expiratory volume in the first second to the forced vital capacity of the lungs; PEF: Peak expiratory flow; 6MWD: Six-minute walking distance.

## Data Availability

Any data related to this study can be provided upon a reasonable request.

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
