# Peer review of "Quality of Life, Family Support, Spirometry, and 6-Minute Walking Distance Differences between COVID-19 and Non-COVID-19 Intensive Care Unit Patients in One Year Following Hospital Discharge"

_healthcare, 2024, doi:10.3390/healthcare12100996_

Round 1

Reviewer 1 Report

Comments and Suggestions for Authors

My comments:

                In general, this is an interesting research. The authors aim to investigate the QoL both COVID-19 and non-COVID-19 patients following ICU discharge, lung function and physical performance and to assess the role of support by family members and friends. They found the QoL of patients, who were discharged from the ICU can be positively affected both by the support they receive from their family environment and friends. However, there are some points should be improved.

1. Abstract

                - The full term of FEV1 and 6-WMT should be provided. Moreover, 6-WMT should be changed to six-minute walk distance (6-MWD).

                - In the conclusion section, you stated that “This study shows that the QoL of patients, who were discharged from the ICU can be positively affected both by the support they receive from their family environment and friends. However, the effect of their family environment and friends on QOL was not analyzed using a regression analysis.  I think that this conclusion section should be change according to your main results.

2. Introduction

            - More studies about the long term impact of ICU required MV on lung function, QOL, and exercise capacity should be mentioned in both COVID and non-COVID. Moreover, rational about the differences of outcomes between survivors with COVID-19 and non-COVID-19 should be mentioned.

3. Materials and Methods

            - The full term off all abbreviations e.g., FEV1, FVC, and SF-36 should be provided before the abbreviation used.   

                - More details about spirometry including device used, standardization used and reference used must be provided.

                - More details about 6-MWT including standardization used must be provided.

                - In subheading 2.4. Respiratory Function Assessment and 6-Min Walk (6MWT), the sentence “The spirometry evaluation was planned to be performed at discharge, 3- months and 12-months follow-up of the patients, using a portable spirometer.” should be changed to past tense.  

                - In the sub heading 2.5. Statistical analysis , please specify paired-sample t- test and non-parametric test, or using Mann-Whitney test and one-way ANOVA was used for?. For example, Mann–Whitney U test was used for comparison continuous data between groups.

4. Results

            - Study flow, this is a prospective observational study, you followed subjects at baseline, month 3, and month 12. However, the number of subjects in each group and reason of loss follow up at each time point was not presented.  Thus, the study flow that includes the number of subjects in each group and reason of loss follow up must be provided.

                - In the table 1, the column that included baseline characteristics of all non-COVID-19 should be provided and 6-MWT should be changed to 6-MWD.

                - In all figures, the decimal should be consistent. Moreover, the p-value that compared across the three groups should be provided.

5. Discussion

            The differences in time points of the follow up period between COVID and non-COVID group should be mention as a limitation of this study. Moreover, there were significance differences in baseline characteristics between groups including ICU stay, APACHE score. However, the regression analysis that included the differences in baseline characteristics between groups was not mentioned.  

Author Response

Larissa, 25/04/2024

Dear Editor,

We were pleased to hear that the Healthcare Journal is interested in a revised version of our manuscript. We are grateful to the editors and the reviewers for the valuable suggestions and comments. After careful consideration, we have adjusted the manuscript accordingly.  All comments have been addressed in the revised version. In more detail:

Below there is a point-by-point description of the changes that have been made to address the editors’ and reviewers' concerns.

EDITOR’S AND REVIEWERS’ COMMENTS (CAPITAL, BLACK)

Answer (lower case, blue)

REVIEVER 1

  1. ABSTRACT

- THE FULL TERM OF FEV1 AND 6-WMT SHOULD BE PROVIDED. MOREOVER, 6-WMT SHOULD BE CHANGED TO SIX-MINUTE WALK DISTANCE (6-MWD).

Thank you for your comment. We have added the full period of FEV1 and 6MWD in abstract (page 1, lines 15-16).

- IN THE CONCLUSION SECTION, YOU STATED THAT “THIS STUDY SHOWS THAT THE QOL OF PATIENTS, WHO WERE DISCHARGED FROM THE ICU CAN BE POSITIVELY AFFECTED BOTH BY THE SUPPORT THEY RECEIVE FROM THEIR FAMILY ENVIRONMENT AND FRIENDS. HOWEVER, THE EFFECT OF THEIR FAMILY ENVIRONMENT AND FRIENDS ON QOL WAS NOT ANALYZED USING A REGRESSION ANALYSIS.  I THINK THAT THIS CONCLUSION SECTION SHOULD BE CHANGE ACCORDING TO YOUR MAIN RESULTS.

Thank you for your comment. We agree that this point should be clarified better. In light of your comment, we have modified the conclusion (page 1, line 17).

  1. INTRODUCTION

- MORE STUDIES ABOUT THE LONG-TERM IMPACT OF ICU REQUIRED MV ON LUNG FUNCTION, QOL, AND EXERCISE CAPACITY SHOULD BE MENTIONED IN BOTH COVID AND NON-COVID. MOREOVER, RATIONAL ABOUT THE DIFFERENCES OF OUTCOMES BETWEEN SURVIVORS WITH COVID-19 AND NON-COVID-19 SHOULD BE MENTIONED.

You are right that this point needs also clarification. In light of your comment, we have modified the introduction (page 1-2, 1-2, line 29-44).

  1. MATERIALS AND METHODS

- THE FULL TERM OFF ALL ABBREVIATIONS E.G., FEV1, FVC, AND SF-36 SHOULD BE PROVIDED BEFORE THE ABBREVIATION USED.   

We have provided the full term off all abbreviations before their use throughout the text.

- MORE DETAILS ABOUT SPIROMETRY INCLUDING DEVICE USED, STANDARDIZATION USED AND REFERENCE USED MUST BE PROVIDED.

Thank you for your remark. We agree that this point should be clarified better. Details have been added in 2.4 Respiratory Function Assessment and 6-Min Walk Distance (6MWD) (page 3, line 120-125).

- MORE DETAILS ABOUT 6-MWT INCLUDING STANDARDIZATION USED MUST BE PROVIDED.

Thank you for your comment. All details about 6MWD were provided in 2.4 Respiratory Function Assessment and 6-Min Walk Distance (6MWD) (page 3, line 129-131, 134-137).

- IN SUBHEADING 2.4. RESPIRATORY FUNCTION ASSESSMENT AND 6-MIN WALK (6MWT), THE SENTENCE “THE SPIROMETRY EVALUATION WAS PLANNED TO BE PERFORMED AT DISCHARGE, 3- MONTHS AND 12-MONTHS FOLLOW-UP OF THE PATIENTS, USING A PORTABLE SPIROMETER.” SHOULD BE CHANGED TO PAST TENSE.  

The sentence was modified based on your remark (page 3, line 119).

- IN THE SUB HEADING 2.5. STATISTICAL ANALYSIS, PLEASE SPECIFY PAIRED-SAMPLE T- TEST AND NON-PARAMETRIC TEST, OR USING MANN-WHITNEY TEST AND ONE-WAY ANOVA WAS USED FOR?. FOR EXAMPLE, MANN–WHITNEY U TEST WAS USED FOR COMPARISON CONTINUOUS DATA BETWEEN GROUPS.

You are right that this point needs also clarification. In light of your comment, we have modified the subheading (2.5 statistical analysis, page 4, line 145-149).

  1. RESULTS

- STUDY FLOW, THIS IS A PROSPECTIVE OBSERVATIONAL STUDY, YOU FOLLOWED SUBJECTS AT BASELINE, MONTH 3, AND MONTH 12. HOWEVER, THE NUMBER OF SUBJECTS IN EACH GROUP AND REASON OF LOSS FOLLOW UP AT EACH TIME POINT WAS NOT PRESENTED.  THUS, THE STUDY FLOW THAT INCLUDES THE NUMBER OF SUBJECTS IN EACH GROUP AND REASON OF LOSS FOLLOW UP MUST BE PROVIDED.

Thank you for your remark. The flow chart of the study was presented at figure 1. In light of your comment, we have modified the figure 1 (page 4).

- IN THE TABLE 1, THE COLUMN THAT INCLUDED BASELINE CHARACTERISTICS OF ALL NON-COVID-19 SHOULD BE PROVIDED AND 6-MWT SHOULD BE CHANGED TO 6-MWD.

In light of this remark, we now present a new column that include baseline characteristics of all non-covid-19 patients (page 5, table 1).

- IN ALL FIGURES, THE DECIMAL SHOULD BE CONSISTENT. MOREOVER, THE P-VALUE THAT COMPARED ACROSS THE THREE GROUPS SHOULD BE PROVIDED.

Thank you for your comment. The figures were modified based on your remark (page 6-9). In light of your comment we now provide the statistic difference between three groups.

  1. DISCUSSION

- THE DIFFERENCES IN TIME POINTS OF THE FOLLOW UP PERIOD BETWEEN COVID AND NON-COVID GROUP SHOULD BE MENTION AS A LIMITATION OF THIS STUDY. MOREOVER, THERE WERE SIGNIFICANCE DIFFERENCES IN BASELINE CHARACTERISTICS BETWEEN GROUPS INCLUDING ICU STAY, APACHE SCORE. HOWEVER, THE REGRESSION ANALYSIS THAT INCLUDED THE DIFFERENCES IN BASELINE CHARACTERISTICS BETWEEN GROUPS WAS NOT MENTIONED.  

Thank you for your comment. Based on your comment we include in the limitations the differences in time points of the follow up period between COVID-19 and non-COVID-19 groups (page 11, line 323-326). Regarding the baseline characteristics of participants, we explained this at discussion (page 10, line 268-269).

Kind regards

Konstantinos Mantzarlis

Doctor in Critical Care Medicine

Reviewer 2 Report

Comments and Suggestions for Authors

Thank you to the authors for the opportunity to review the article, and I hope these revisions can help improve it.

The abstract has limitations, starting with poor comprehensibility due to numerous unexplained acronyms. Moreover, in the abstract, the numerical representation of results needs coherence, particularly regarding decimal points in means and standard deviations. Statistical significance is indicated as p<0.0001, which perhaps was intended to be 0.001. Additionally, the mention of the "mental domain" in the final results section of the abstract lacks reference to a specific scale.

The introduction is highly interesting and well-written. However, several acronyms are introduced before they are explained (see line 28). Also, throughout the manuscript, "COVID-19" is inconsistently capitalized. Please ensure uniformity throughout.

Surprisingly, the authors discussed potential physical and psychological consequences in patients but did not address falls, a major global health concern, or strategies adopted by healthcare systems (see DOI: https://doi.org/10.1177/25160435241246344). Several bibliographic citations need to be included in the introduction section (see lines 32, 45, 46).

In the methods section, there are discrepancies regarding the period of interest compared to what is stated in the abstract. Please provide consistent information. Paragraph 2.3 is unclear and confusing; please clarify the questionnaire in a more organized manner.

Is Figure 1 essential to the study? Regarding Table 1, only results with p<0.05 are mentioned; are there no results with p<0.001? Which statistical tests were used? The use of the APACHE score needs to be adequately explained in the Materials and Methods section.

The descriptions of Figures 3 and 4 need to be completed and adequate; they need to be explanatory.

Why start the discussion with a list of critical findings? The discussions are scant compared to the presented results, which could have been more organized. Please present the results more organized and logically and discuss them adequately, focusing on the latest scientific evidence. 

The manuscript needs adequate bibliographic research.

In light of these changes, please revise the conclusions and abstract accordingly.

Best regards.

Author Response

Larissa, 25/04/2024

Dear Editor,

We were pleased to hear that the Healthcare Journal is interested in a revised version of our manuscript. We are grateful to the editors and the reviewers for the valuable suggestions and comments. After careful consideration, we have adjusted the manuscript accordingly.  All comments have been addressed in the revised version. In more detail:

Below there is a point-by-point description of the changes that have been made to address the editors’ and reviewers' concerns.

EDITOR’S AND REVIEWERS’ COMMENTS (CAPITAL, BLACK)

Answer (lower case, blue)

REVIEWER 2

  1. THE ABSTRACT HAS LIMITATIONS, STARTING WITH POOR COMPREHENSIBILITY DUE TO NUMEROUS UNEXPLAINED ACRONYMS. MOREOVER, IN THE ABSTRACT, THE NUMERICAL REPRESENTATION OF RESULTS NEEDS COHERENCE, PARTICULARLY REGARDING DECIMAL POINTS IN MEANS AND STANDARD DEVIATIONS. STATISTICAL SIGNIFICANCE IS INDICATED AS P<0.0001, WHICH PERHAPS WAS INTENDED TO BE 0.001. ADDITIONALLY, THE MENTION OF THE "MENTAL DOMAIN" IN THE FINAL RESULTS SECTION OF THE ABSTRACT LACKS REFERENCE TO A SPECIFIC SCALE.

Thank you for your comment. The abstract was modified based on your remark (page 1, line 7-17).

  1. THE INTRODUCTION IS HIGHLY INTERESTING AND WELL-WRITTEN. HOWEVER, SEVERAL ACRONYMS ARE INTRODUCED BEFORE THEY ARE EXPLAINED (SEE LINE 28). ALSO, THROUGHOUT THE MANUSCRIPT, "COVID-19" IS INCONSISTENTLY CAPITALIZED. PLEASE ENSURE UNIFORMITY THROUGHOUT.

Thank you for your remarks. Changes have been made throughout the text according to your comment.

  1. SURPRISINGLY, THE AUTHORS DISCUSSED POTENTIAL PHYSICAL AND PSYCHOLOGICAL CONSEQUENCES IN PATIENTS BUT DID NOT ADDRESS FALLS, A MAJOR GLOBAL HEALTH CONCERN, OR STRATEGIES ADOPTED BY HEALTHCARE SYSTEMS (SEE DOI: HTTPS://DOI.ORG/10.1177/25160435241246344). SEVERAL BIBLIOGRAPHIC CITATIONS NEED TO BE INCLUDED IN THE INTRODUCTION SECTION (SEE LINES 32, 45, 46).

You are right. The study did not address falls and therefore with added that as a limitation (page 11, line 327-329). Furthermore, based on your comment we include several bibliographic citations in the introduction (page 1-2, line 29-44).

  1. IN THE METHODS SECTION, THERE ARE DISCREPANCIES REGARDING THE PERIOD OF INTEREST COMPARED TO WHAT IS STATED IN THE ABSTRACT. PLEASE PROVIDE CONSISTENT INFORMATION. PARAGRAPH 2.3 IS UNCLEAR AND CONFUSING; PLEASE CLARIFY THE QUESTIONNAIRE IN A MORE ORGANIZED MANNER.

In light of your comment, we modified the methods section (page 2, line 66). Also, we have modified the subheading 2.3 Questionnaire interview, page 3, line 94-116).

  1. IS FIGURE 1 ESSENTIAL TO THE STUDY? REGARDING TABLE 1, ONLY RESULTS WITH P<0.05 ARE MENTIONED; ARE THERE NO RESULTS WITH P<0.001? WHICH STATISTICAL TESTS WERE USED? THE USE OF THE APACHE SCORE NEEDS TO BE ADEQUATELY EXPLAINED IN THE MATERIALS AND METHODS SECTION.

Thank you for your comment. We modified the figure 1 to show the distribution of patients who participated in the study. The statistical tests that were used are described in Methods 2.5 Statistical Analysis (page 4, line 145-151). In addition, you are right that APACHE II score needs also explanation. So, we describe the APACHE II score at 2.2. Data collection (page 2, line 82-85).

  1. THE DESCRIPTIONS OF FIGURES 3 AND 4 NEED TO BE COMPLETED AND ADEQUATE; THEY NEED TO BE EXPLANATORY.

Thank you for your comment. We describe the figure 3 (page 7, line 219-220) and figure 4 (page 8, line 238-239).

  1. WHY START THE DISCUSSION WITH A LIST OF CRITICAL FINDINGS? THE DISCUSSIONS ARE SCANT COMPARED TO THE PRESENTED RESULTS, WHICH COULD HAVE BEEN MORE ORGANIZED. PLEASE PRESENT THE RESULTS MORE ORGANIZED AND LOGICALLY AND DISCUSS THEM ADEQUATELY, FOCUSING ON THE LATEST SCIENTIFIC EVIDENCE. 

Thank you for your comment. We considered that was interesting to perform the main findings in the start of the discussion. In light of your comment, we modified the discussion (page 10-11, line 260-267, 268-269, 275-277, 278-282, 289-291, 292-296, 307-308).

  1. THE MANUSCRIPT NEEDS ADEQUATE BIBLIOGRAPHIC RESEARCH.

Thank you for your comment. New references were added.

  1. IN LIGHT OF THESE CHANGES, PLEASE REVISE THE CONCLUSIONS AND ABSTRACT ACCORDINGLY.

Thank you for your comment. Abstract and Conclusions were revised. Please refer to the specific parts of the manuscript.

Kind regards

Konstantinos Mantzarlis

Doctor in Critical Care Medicine

Round 2

Reviewer 1 Report

Comments and Suggestions for Authors

All of my comments have been addressed by authors. This manuscript can be accepted as a current form. 

Author Response

Larissa, 3/05/2024

Dear Editor,

We were pleased to hear that the Healthcare Journal is interested in a revised version of our manuscript. We are grateful to the editors and the reviewers for the valuable suggestions and comments. After careful consideration, we have adjusted the manuscript accordingly.  All comments have been addressed in the revised version.

Kind regards

Konstantinos Mantzarlis

Doctor in Critical Care Medicine

Reviewer 2 Report

Comments and Suggestions for Authors

I want to congratulate you on your efforts to address the review comments and implement the revisions in the manuscript, but nevertheless, I believe that you should re-evaluate the methodology and results in more depth before re-submitting the manuscript.

Author Response

Larissa, 3/05/2024

Dear Editor,

We were pleased to hear that the Healthcare Journal is interested in a revised version of our manuscript. We are grateful to the editors and the reviewers for the valuable suggestions and comments. After careful consideration, we have adjusted the manuscript accordingly.  All comments have been addressed in the revised version. In more detail:

Below there is a point-by-point description of the changes that have been made to address the editors’ and reviewers' concerns.

EDITOR’S AND REVIEWERS’ COMMENTS (CAPITAL, BLACK)

Answer (lower case, blue)

Based on your comment and for a better explanation and interpretation of the results, changes were made to the discussion to give more depth to the results (lines 264-273, 296-299, 302-306, 308-332, 335-344).

Kind regards

Konstantinos Mantzarlis

Doctor in Critical Care Medicine
